# Global Cutaneous Mucormycosis: A Systematic Review

**DOI:** 10.3390/jof8020194

**Published:** 2022-02-16

**Authors:** Anna Skiada, Maria Drogari-Apiranthitou, Ioannis Pavleas, Eirini Daikou, George Petrikkos

**Affiliations:** 1First Department of Medicine, Laiko Hospital, National and Kapodistrian University of Athens, 11527 Athens, Greece; eirini.daikou@gmail.com; 2Fourth Department of Internal Medicine, General University Hospital “Attikon”, National and Kapodistrian University of Athens, 12462 Athens, Greece; mdrogari@hotmail.com (M.D.-A.); g.petrikkos@euc.ac.cy (G.P.); 3Intensive Care Unit, Laiko Hospital, 11527 Athens, Greece; ipavleas@yahoo.com; 4School of Medicine, European University of Cyprus, Engomi 2404, Cyprus

**Keywords:** mucormycosis, cutaneous, zygomycosis, review, epidemiology, trauma

## Abstract

Cutaneous mucormycosis is the third most common clinical type of mucormycosis. The signs and symptoms vary widely, and it is important to make the diagnosis as early as possible in order to achieve a better outcome. We present a systematic review of its epidemiology, clinical presentation, diagnosis, and treatment, analyzing cases published from 1958 until 2021. The review was conducted according to the PRISMA guidelines and included 693 cases from 485 articles from 46 countries. Most publications were from North America (256 cases, 36.9%) and Asia (216 cases, 31.2%). The most common risk factors were diabetes mellitus (20%) and hematological malignancies (15.7%). However, a large proportion of published cases (275, 39.6%) had no identified underlying disease. The most common mode of transmission was trauma (54%), and 108 (15.6%) cases were healthcare-associated. In this review, 291 (42.5%) patients had localized infection, and 90 (13%) had disseminated mucormycosis. In Europe, N. America and S. America, the most common genus was *Rhizopus* spp., while in Asia it was *Apophysomyces* spp. (34.7%). Treatment was performed with antifungals, mainly amphotericin B, and/or surgery. Mortality was significantly lower when both antifungals and surgery were applied (29.6%).

## 1. Introduction

Mucormycosis is an emerging fungal infection, caused by fungi of the order Mucorales. Diabetes mellitus is the most common underlying disease globally [1,2]. Other risk factors include hematological and other malignancies, transplantation, prolonged neutropenia, corticosteroids, trauma, burns, iron overload, illicit intravenous drug use, or neonatal prematurity [3]. The increasing prevalence of diabetes, especially in low- and middle-income countries has resulted in the increased incidence of mucormycosis [4]. An increased incidence has also been reported in Western Europe and the USA, as new immunomodulating agents are used in the treatment of cancer and autoimmune diseases, and as more transplantations are performed [5]. Cutaneous mucormycosis is the third most common clinical type of this infection, after rhinocerebral and pulmonary [3]. Its epidemiology is evolving, as new diagnostic tools lead to the identification of previously uncommon species, such as *Saksenaea erythrospora* [6] and *Apophysomyces mexicanus* [7]. A large proportion of patients with cutaneous mucormycosis have no underlying disease, and their infection is a result of trauma. The signs and symptoms vary widely, and it is important to make the diagnosis as early as possible in order to achieve a better outcome. For this reason, we performed a systematic review of the epidemiology, clinical presentation, diagnosis, and treatment of cutaneous mucormycosis, analyzing case reports and case series published from 1958 until 2021.

## 2. Methods

The review was conducted according to the PRISMA (Preferred Reporting Items for Systematic reviews and Meta-Analyses) guidelines for systematic reviews [8].

### 2.1. Search Strategies and Eligibility Criteria

Cases were identified from PubMed, using the keywords “cutaneous”, “mucormycosis”, “zygomycosis”, “soft tissue”, “fasciitis”, “trauma”, “ulcer”, “Rhizopus”, “Mucor”, “Lichtheimia”, “Apophysomyces”, “Saksenaea”, “Syncephalastrum”, and “Cunninghamella”. The references of the relevant articles were also searched. Only articles published in English, and pertaining to humans, were included. The cases had to have documentation of: (i) age and sex of the patient, (ii) underlying diseases/predisposing factors, (iii) site(s) of infection, (iv) methods of diagnosis, (v) description of the applied treatment, and (vi) outcome. Twelve patients who left against medical advice were also included. Selected articles presented either primary or secondary/disseminated cutaneous mucormycosis. In the latter case, we included only those where the diagnosis of mucormycosis was initially made from the cutaneous lesion(s). Diagnoses had to be made by histopathology and/or the identification of the responsible fungus, either by culture, or by molecular methods.

The first case identified by this search strategy was published in 1958. The final search was published in July 2021.

Publications of case series with no description of individual patient data, review articles, conference abstracts, or editorials, and those with poorly described cases, were excluded.

### 2.2. Data Collection Process and Data Items

Collected data were inserted in an Excel database. Data were extracted independently by three authors (A.S., M.D.A., and E.D.), and any discrepancies were discussed with I.P. and G.P. The data items were age, sex, country of origin, year of publication, underlying disease/predisposing factors, mode of transmission, site of infection, clinical presentation, methods of diagnosis (including biopsy, autopsy, direct microscopy, cultures, and molecular methods), modes of treatment, and outcome (all-cause mortality). Clinical presentation was classified as localized, with deep extension, or disseminated, depending on the extent of the fungal invasion [3].

### 2.3. Statistical Analysis

Descriptive statistics were conducted using the Statistical Package for the Social Sciences for Windows (version 11.5; SPSS Inc., Chicago, IL, USA). Categorical variables are reported as percentages and continuous variables (except for years of publication) as median and range.

## 3. Results

As described in the PRISMA diagram (Figure 1), after screening the retrieved articles, 693 cases from 484 articles were included in the analysis. The oldest identified case was published in 1958. The number of cases per year are shown in Figure 2. The data included were from 46 countries. The geographic distribution of cases is shown in Figure 3. Most publications were from North America (256 cases, 36.9%) and Asia (216 cases, 31.2%). The rest were: 149 from Europe (21.5%), 29 from Central and South America (4.2%), 40 from Australia (5.8%), and 3 from Africa (0.4%). The list of articles analyzed is presented in the Appendix A.

### 3.1. Demographics

Of the 693 patients, 130 (19%) were younger than 18 years of age. Of these 130, 93 were older than one year (median age 11 years, range 1–17 years), 12 were between 1 and 12 months (median age 6 months, range 1.7–11 months), and 25 were neonates (median age 10 days, range 1 to 22 days). The median age of the patients ≥18 years old was 45 years (range 18 to 90). The ratio of female to male patients was 1/1.8 (247/446).

### 3.2. Underlying Diseases and Mode of Transmission

The underlying diseases/conditions are shown in Table 1. The most common risk factors were diabetes mellitus (20%) and hematological malignancies (15.7%). However, a large proportion of published cases (275, 39.6%) had no identified underlying disease. Of these 275 cases, 239 (87%) developed mucormycosis as a result of trauma or burns. The remaining 36 cases with no underlying diseases and unknown modes of transmission were mostly due to *Mucor irregularis* [9], *Apophysomyces* spp. [10], or *Saksenaea* spp. [11], with a prolonged clinical history, ranging from 5 weeks to 17 years, in 72% of them.

The reported modes of transmission are shown in Table 2. The majority (54%) were trauma, of varying magnitude, from as small as an insulin injection [12] to as large as that resulting from a tornado [13] or from a war injury [14,15]. A significant number of cases (108, 15.6%) were healthcare-associated, while 203 (29.3%) had no known mode of transmission.

### 3.3. Clinical Presentation

Cutaneous mucormycosis is characterized as localized when only the skin and the subcutaneous tissue are involved, as having deep extension when the infection invades underlying muscle, tendons, or bone, and as disseminated when it affects other, noncontiguous organs. In this review, 291 (42.5%) patients had localized infections, 304 (44.5%) had deep extension of the disease, and 90 (13%) had disseminated mucormycosis. In the majority of cases with disseminated disease, that is, in 67/90 (74%), the portal of entry for the fungus was considered to be the skin, while in 16/90 (18%), the mycosis disseminated to the skin from an internal organ, mainly from the lung. In seven cases (8%), the portal of entry was not clear because the diagnosis of cutaneous and pulmonary mucormycosis was made simultaneously.

The upper and lower extremities were the most common sites of cutaneous mucormycosis, but any area of the skin could be affected (Figure 4).

The clinical manifestations of cutaneous mucormycosis vary. In the present review, 70 (10%) cases did not have a detailed description of signs and symptoms, so the analysis was completed for 623 cases. The classical clinical sign of mucormycosis is a black eschar. However, not all cases present in the same way, and, quite often, the initial presentation is different. In 346 (55%) cases, the words “necrosis”, “necrotic”, or “necrotizing” were used to describe the cutaneous lesions. In 59 (9%) cases, the word “eschar” was used, and in another 10 cases, the phrase “blackish discoloration” of the skin, the wound, or the ulcer was used. The infection was described as “cellulitis” in 20 cases and as “abscess” in seven. Before the full-blown picture of necrosis emerged, in many cases, the initial presentation was a small nodule [16], erythematous nodules [17], ulcerative, nodular wounds [18], vesicles grouped in a circular fashion [19], an ulcer with a white mold-like growth after a spider bite [20], a cottony growth [21], a small blister [22], tender, purpuric macules with peripheral erythema [23], a 1-cm hard papule [24], a vesicular eruption [25], an indurated mass [26], or an indurated, woody, and hard swelling [27]. In two cases, the infection presented as a “purple bull’s-eye lesion” [28] or ‘‘bull’s-eye infarct” [29]. In another three, the lesions were described as ecthyma gangrenosum [30], “erythema nodosum-like rash” [31], or “pyoderma gangrenosum-like” [32]. Pain severity varied. The lesions were completely painless in some cases, and painful or very painful in others. The patients could also have signs and symptoms of sepsis, depending on the underlying disease and the extent of the infection.

In the majority of cases, the infection progressed rapidly, sometimes leading to gangrene and hematogenous dissemination. However, there are several case reports, mainly from China, presenting cases of mucormycosis due to *Mucor irregularis*, where the infection progressed very slowly [33]. In addition to *M. irregularis*, cases due to *Saksenaea vasiformis* have been reported to progress over months [11,34]. There have also been reports of slowly progressing lesions due to *Mucor hiemalis* [35,36], *Syncephalastrum* sp. [37], and *Rhizopus microsporus* [38,39].

### 3.4. Diagnosis/Infecting Organisms

Of the 693 reported cases, 563 were diagnosed by both histopathology and culture, another 11 with histopathology and tissue PCR, and 6 were diagnosed post-mortem, with autopsy and culture. Therefore, in total, 580 (83.6%) of the published cases were proven. Biopsies were performed in 658 cases (95%), direct microscopy in 214 (31%), and cultures in 645 (93%). The cultures were positive in 595 of 645 cases (92%), and a mucoralean fungus was identified at the genus- or species-level in a total of 605 cases. In further 11 cases with negative or not-performed cultures, mucoralean species were identified by PCR and sequencing. The isolated fungi are shown in Table 3, and their geographic distribution in Figure 5.

There were several cases where more than one fungus was isolated. In one case of disseminated mucormycosis with a hematological malignancy and HSCT, *R. microsporus* and *Lichtheimia* spp. were isolated [40]. In another two patients with neutropenia due to hematological malignancies, *Aspergillus* spp. was isolated in addition to *Mucor circinelloides* in one [41] and *Rhizopus rhizopodiformis* in the other [42]. In an immunocompetent patient who had a farm-related accident, *Lichtheimia* sp., *Rhizopus* sp., and *Aspergillus* sp. were isolated [43], while in another patient who was in a traffic accident, *Rhizomucor* sp. and *Fusarium* sp. were found [44]. Sequencing was performed in 158 cases, MALDI-TOF in eight, tissue PCR in 14, and pan-fungal PCR in four cases.

### 3.5. Treatment and Outcome

The patients received antifungal medication and/or surgery. Surgery consisted either of local debridement or amputation of an infected limb. Antifungals were given to 610 (88%) patients and surgery was performed on 537 (77%). In 483 (70%) cases, treatment included both antifungals and surgery. The drug used more frequently was an amphotericin B formulation (560 cases, 81%). More specifically, liposomal amphotericin B was given in 262 (38%) cases, amphotericin B lipid complex in 18 (2%), amphotericin B deoxycholate in 238 (34%), and unspecified amphotericin B in 42 (6%), isavuconazole in 10 (1%), posaconazole in 85 (12%), and caspofungin in 10 (1%), either alone or in combination.

Crude mortality was 34%, excluding the 12 patients who left against medical advice. The mortality of neonates (aged < 1 month) was 48%. Ninety-two percent of these neonates were premature. Mortality according to clinical presentation was 18% for localized, 39% for deeply extended, and 70% for disseminated infection. As shown in Table 3, mucormycosis due to *Cunningamella* spp. was associated with a significantly higher mortality (67%), while with the other genera mortality ranged from 20% to 42%. The mortality of the 54 patients who were treated only with surgery was 46.2%; of the 127 treated only with antifungals, the mortality was 36%, and the mortality of the 483 treated with both antifungals and surgery was 29.6%.

## 4. Discussion

Cutaneous mucormycosis is the third most common type of mucormycosis, following rhino-orbito-cerebral and pulmonary. We present here an update of its epidemiology, clinical features, predisposing risk factors, diagnosis, and treatment. We performed a systematic review, according to the PRISMA guidelines, and found 693 eligible cases, published from 1958 to 2021. There are publications reporting increased incidence of mucormycosis in general, as a result either of an increased prevalence of diabetes, especially in Asia, or of new treatments for malignancies and autoimmune diseases [4,5]. However, it is not possible to estimate the exact incidence or prevalence of cutaneous mucormycosis, because most data are from case reports or case series. Most cases included in the present review were from the USA (249) and India (118). There are multiple factors leading to increased numbers of mucormycosis in a country. There may actually be high incidence, or an increased rate of recognition because of better awareness, expertise, and competence in mycological diagnosis [45]. From 1958 to 2000, there were only 8 cases of cutaneous mucormycosis published from India, and 134 from the USA. As the diagnostic facilities in healthcare centers of developing countries are improving, the number of cases reported, especially from India, are increasing alarmingly. COVID-19 is another factor contributing to the increase of cases of mucormycosis, mainly in India and neighboring countries [46]. In a study by Patel et al., during the first wave of COVID-19 infection, a 2.1-times increase in mucormycosis cases was observed, compared to the previous year [47]. However, the most common clinical presentation of Coronavirus disease-associated mucormycosis (CAM) is rhino-orbital, and cutaneous cases are very rare [47,48,49,50]. In the current review, there was only one case of CAM in a patient with diabetes mellitus, who had recently undergone heart transplantation [51].

The principal risk factor for mucormycosis in India is uncontrolled diabetes mellitus [4]. In the present review, diabetes mellitus was globally reported in 20% of cases, being, thus, the most common underlying disease. In the cases from the USA, diabetes was present in 16.4%, while in the ones from India, it was present in 24.5%. Hematological malignancies were the second most common underlying disease, present in 15.7% of cases. In the USA, they were the main risk factor in 18.4% of cases, while in India, they were so only in 4.2% of cases. The trend in Europe was similar to that in the USA. An important finding is that in 39.6% of published cases, the patients were immunocompetent. The number was higher in cases from India, where 58.4% had no identified underlying disease, as well as in cases from China (57%) and Australia (60%). In neonates, the most common condition predisposing to cutaneous mucormycosis was prematurity and extremely low birth weight, as has been previously reported in the literature [52].

Cutaneous mucormycosis develops when there is a disruption of the normal protective cutaneous barrier. This is usually the result of trauma, which leads to the direct inoculation of spores into the skin. When the host has no underlying diseases and the trauma is minor, the infection may be contained if diagnosed early. However, when the patient has sustained major trauma and there are recurring episodes of sepsis, shock, and multiple blood transfusions, an immune-deficient state emerges, predisposing to the development of mucormycosis [53]. Furthermore, if the host is immunocompromised, even a minor trauma may lead to dissemination. In the current review, trauma—including burns—was the mode of transmission in 54.1% of cases (Table 2). A large proportion was due to motor vehicle accidents (104/375, 27.7%). In a case series by Ingram et al. [54], the proportion of cases with an MVA-associated trauma with no other risk factors was 78%, a number significantly higher than that published in older reviews (3%, 20%) [1,55]. The authors stated that all of these cases involved either unrestrained passengers or motorcycles, and hypothesized that this may have resulted in higher direct environmental trauma and, hence, greater opportunity for wound contamination with Mucorales [54]. Another complicating factor in trauma from MVAs, leading to increased mortality, are burns [54,56,57].

Other cases of major trauma were due to natural disasters. Eight cases of necrotizing fasciitis because of mucormycosis were diagnosed in 1985, when a volcanic eruption of major proportions wiped out the town of Armero in Colombia [58]. They were not included in this review, because there was no description of individual patient data. The patients suffered major injuries progressing rapidly to extensive lesions. Six of them died despite radical debridement and amputations. In four of them, the fungus was identified as *R. arrhizus*. Only two received amphotericin B, but the diagnosis was delayed [58]. Three more reported cases emerged after the tsunami in the Indian Ocean in 2004 [59,60,61]. Two were due to *Apophysomyces elegans* and survived after extensive debridement, intensive care, and antifungal treatment [59,60]. A third one had mucormycosis with *Mucor* sp. [61]. A cluster of 13 cases of necrotizing cutaneous mucormycosis was also reported among the survivors of the tornado in Joplin, Missouri in 2011. The infections were caused by *Apophysomyces trapeziformis*, following lacerations and penetrating injuries from airborne material, including soil, gravel, wood, and glass. Mortality was 38% [13]. It must be stressed that infections in survivors of natural disasters are multimicrobial, with most patients suffering bacterial infections. Mycoses could be, thus, under-diagnosed, given also their similarity to bacterial infections, particularly during the early stages of infection [62]. With climate change and global warming contributing to greater frequency and severity of extreme weather events [63], the numbers of disaster-associated fungal infections, mucormycosis included, could increase. Therefore, efforts should be made to increase the index of suspicion. Histopathology examinations are crucial for early diagnosis and the prompt initiation of the appropriate treatment in these patients.

Combat-related injuries are another type of major trauma predisposing to mucormycosis and other invasive fungal infections (IFIs). In the case series published by Warkentien et al. [64], presenting IFIs in US military personnel injured in Afghanistan, Mucorales were cultured from 16 (43%) of the 37 analyzed cases. In 28% of cases, multiple mold species were isolated from the infected wounds. Common findings included blast injury during foot patrol (100%), with traumatic lower extremity amputation, and large-volume blood transfusions. Rodriguez et al. [65] showed in a multivariate analysis that these were the major risk factors for IFIs in combat-related trauma.

Mucorales are ubiquitous in nature, and spores can be found in soil, compost piles, fruits, and decaying organic material, so accidents occurring in farms, or during gardening, may lead to mucormycosis. Minor trauma, such as bites from insects or animals, have also been implicated in the transmission of the infection. In eight published cases, mucormycosis was the result of immersion in freshwater: in three of them, the patient had been swimming in a river or brackish water, and did not recall any trauma [66,67,68]; in another two, there was a history of small lacerations sustained in water [69,70], while in the other three, there was a history of a severe sailing accident in a river [71], near drowning in a polluted bayou [72], and immersion in freshwater after extensive burning in the third one [73].

Most mucormycosis cases are community-acquired. However, nosocomial acquisition is very important. In our review, 15.6% of cases were healthcare-associated, mainly from adhesive tapes or bandages, and intravenous or arterial catheters. Surgical wounds may also be complicated by mucormycosis, either directly or due to contaminated dressings. Intramuscular or even subcutaneous injections are another source of infection by Mucorales. A relatively large number has been published by Chander et al. [6,74,75], who reported 12 such cases. Of those, only three had underlying diabetes mellitus, and 50% died because of necrotizing fasciitis. Outbreaks have also been linked to contaminated hospital linens [40,76]. Burn patients hospitalized in intensive care units are particularly susceptible to fungal infection because of the loss of the skin barrier and acquired immune deficiency. In a single-center study in a specialized referral burn unit, mortality for patients with mucormycosis was 62% [77].

The clinical diagnosis of cutaneous mucormycosis has a low sensitivity and specificity because the necrotic eschar may not be present and its absence does not preclude the diagnosis [78]. A small blister or nodule may be overlooked, or, in the case of necrotizing fasciitis, multiple bacteria may be isolated. The diagnosis may be challenging in burn patients due to atypical wound aspects, which are the result of skin excision and grafting. In addition, there are cases of mucormycosis, mainly due to *M. irregularis*, where the infection progresses very slowly, in contrast to the classical presentation of mucormycosis. Xia ZK et al. [33] describes a case where the patient “developed a localized inflammatory lesion 14 years ago, following the excision of a pea-sized red tubercle on the right paranasal area” (*sic*) and finally “developed progressive destruction of the nasal sinus, nasal septum, soft and hard palates, and uvula.” A high index of suspicion is, therefore, required. In cases of minor or major trauma, even if the patient is immunocompetent, mucormycosis should be included in the differential diagnosis. Direct microscopy of KOH wet mounts provides a rapid presumptive diagnosis, and it is strongly recommended, along with histopathology, by a panel of experts of the European Confederation of Medical Mycology in cooperation with the Mycoses Study Group Education and Research Consortium (ECMM/MSG ERC) [79]; however, in our review, it was performed in only 31% of cases. Definitive diagnosis is made with the combination of histopathology and culture. Histopathology is very important, because it shows that the infection is invasive. For example, in a case of a cirrhotic patient who fell on a crab trap and developed cellulitis, both *A. elegans* and *Fusarium* sp. were isolated, but histopathology revealed that *Fusarium* was just a colonizer and was not invading the tissues [80]. However, even when fungal hyphae are seen in histopathologic analysis, fungal cultures are positive in only 50% of cases [81]. In the current study, cultures were performed in 93% of cases, and 92% of those were positive. This is probably due to publication bias.

As shown in Figure 5, there was a significant geographic variation of Mucorales organisms causing cutaneous mucormycosis. In Europe, N. America, and S. America, the most common genus was *Rhizopus* spp. Of the 61 cases due to *Lichtheimia* spp., 40 (65.5%) were from Europe. This is consistent with findings from other studies [2,82]. In Asia, the most commonly isolated genus was *Apophysomyces* spp. (34.7%), followed by *Rhizopus* spp. [24%] and *Saksenaea* spp. [18%]. In several publications, it has been reported that *Apophysomyces* spp. and *Saksenaea* spp. can initiate disease in apparently normal hosts, following penetrating trauma during accidents in tropical and sub-tropical areas [45,83]. In the analysis of 122 cases of posttraumatic mucormycosis from the literature by Lelievre et al. [84], *S. vasiformis* and *A. elegans* complex were responsible for 9% and 18.9% of cases, respectively. In India, *A. elegans* is the second most common causative agent, after *R. arrhizus* [45]. In the current review, 33% of cases from Australia were due to *Saksenaea* spp., 28% to *Rhizopus* spp., and 20.5% to *Apophysomyces* spp., while in America and Europe, *Rhizopus* spp. was the most commonly isolated genus. Another species, which was found almost exclusively in cases from Asia (China and India), was *Mucor irregularis*. It should be noted, however, that several of the cases of *Rhizomucor* spp. reported from Europe, Asia, and N. America were in fact *M. irregularis*. *Rhizomucor variabilis* was renamed as *M. irregularis* because results of rDNA internal transcribed spacer (ITS) sequencing showed closer phylogenetic distance to *Mucor* species, especially *M. hiemalis* [85,86]. As molecular-based assays are evolving, they are used more often to aid in the detection or the identification of Mucorales.

Rapid diagnosis of mucormycosis and the early initiation of treatment are of paramount importance. Management consists of the reversal of risk factors, if possible, antifungal therapy, and surgical treatment. First-line treatment with high-dose liposomal amphotericin B is strongly recommended [79]. Amphotericin B deoxycholate, however, may be the only option in resource-limited settings. In our review, liposomal amphotericin B was reportedly administered in 38% of cases, and amphotericin B deoxycholate in 34%, mainly in older cases or in those from developing countries. In newer publications, there are also reports of using posaconazole or isavuconazole. In cutaneous mucormycosis, surgical treatment is much more commonly feasible than with the other types of mucormycosis. In the “RetroZygo” study from France, more patients (94%) underwent surgery for posttraumatic mucormycosis than did those with other forms (48%), and survival at day 90 was greater in post-traumatic mucormycosis (88%), in comparison to that of other types of mucormycosis (48%) [87,88]. In an older publication by Adam et al. [89], disease-related mortality among patients with involvement of only an extremity was 15.5%, while the combined mortality associated with disease of the other locations was 32%. In the current review, the crude mortality of patients with mucormycosis of extremities and other locations was 26% and 43%, respectively. The lower mortality associated with disease of the extremities may be due to the fact that complete debridement of the infected tissue is more easily performed in these patients. Surgical debridement consists of a complete resection of necrotic tissue, with a careful re-evaluation of the wound, in order to diagnose the remaining infection. Multiple debridements are routine, and there have been reports of 19 or 20 repeat debridements, or a final amputation [90]. McDermott et al. proposed calcofluor fluorescence microscopy as a method for guiding the extent of intraoperative surgical debridement [91]. Kyriopoulos et al., reporting from a department of plastic surgery and burns, suggested that, in addition to systemic antifungal therapy, the local application of impregnated dressings on the wound bed should be used, and stressed that the plastic surgeon must be patient and wait for negative swab cultures and biopsies prior to reconstruction [92].

In the current review, mortality was significantly lower when both antifungals and surgery were applied. This is consistent with previous publications [82]. However, even in cases of localized infection, mortality was 18%, which is not negligible. Efforts should be made to optimize diagnosis and treatment of this potentially lethal fungal infection.

### Strength and Limitations

The current review is the most recent and largest analysis of the epidemiology, diagnosis, and treatment of cutaneous mucormycosis. It is a systematic review, and stringent inclusion criteria were applied. However, it is possible that some cases may have been overlooked. Furthermore, the review was retrospective, and not designed to search specifically for mortality or any other parameters of the disease. In addition, publication bias is a limitation, because case reports of unusual or atypical cases are more likely to be published.

## Figures and Tables

**Figure 1 jof-08-00194-f001:**
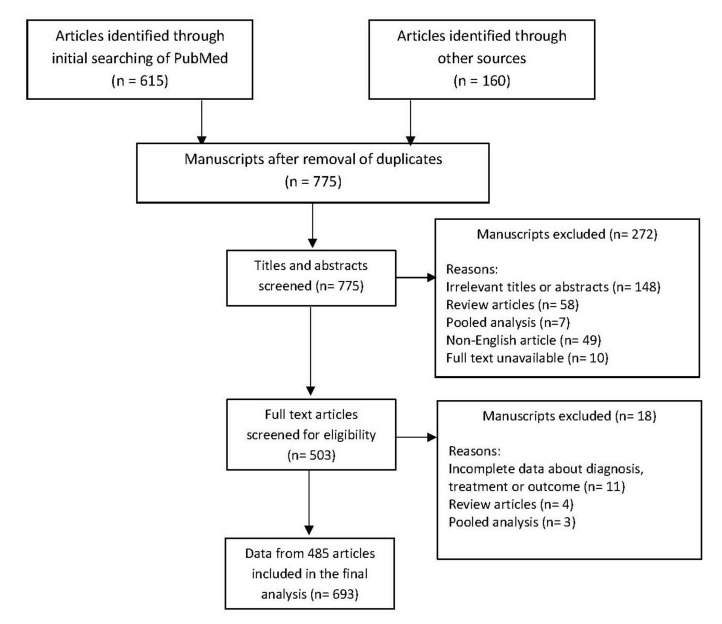
PRISMA diagram describing the identification of eligible cases.

**Figure 2 jof-08-00194-f002:**
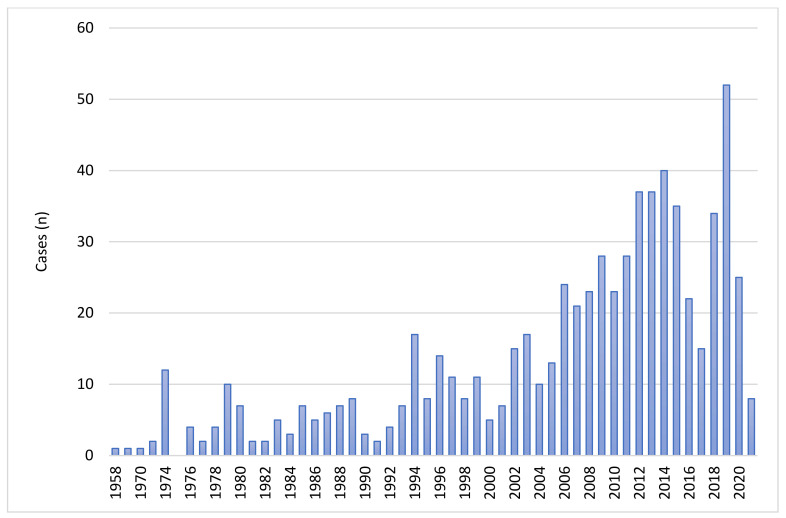
Cases of cutaneous mucormycosis per year from 1958 to 30 June 2021.

**Figure 3 jof-08-00194-f003:**
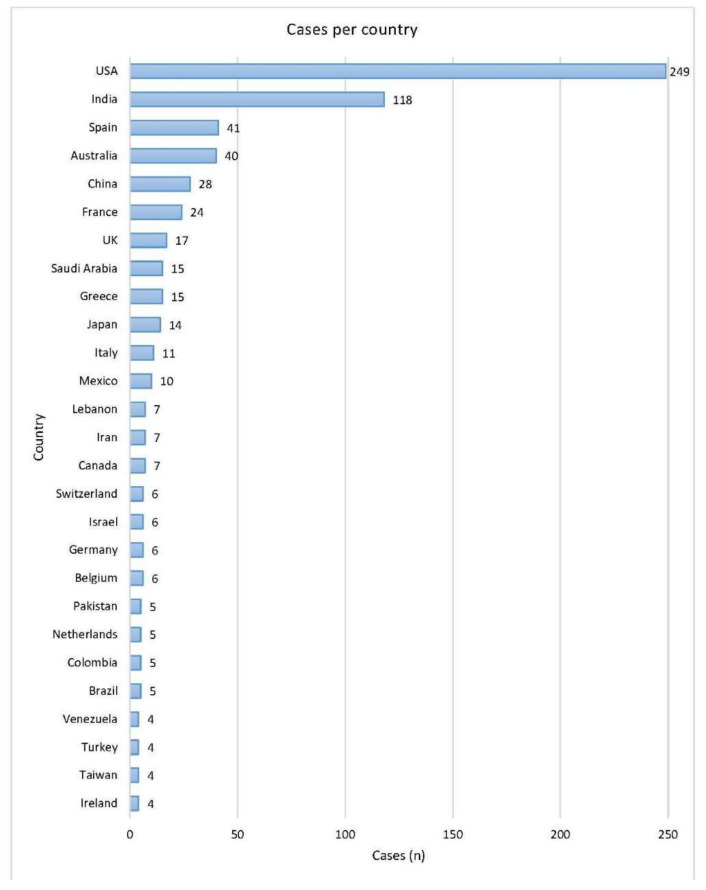
Geographic distribution of cases. Countries in the figure had at least four cases. The rest were: Thailand, Qatar, and Argentina had three cases each; Tunisia, Singapore, S. Korea, Poland, and Czech Republic had two cases each; Sweden, Portugal, Finland, Lithuania, Guatemala, Ecuador, Sri Lanka, Oman, Nepal, Kuwait, and South Africa had one case each.

**Figure 4 jof-08-00194-f004:**
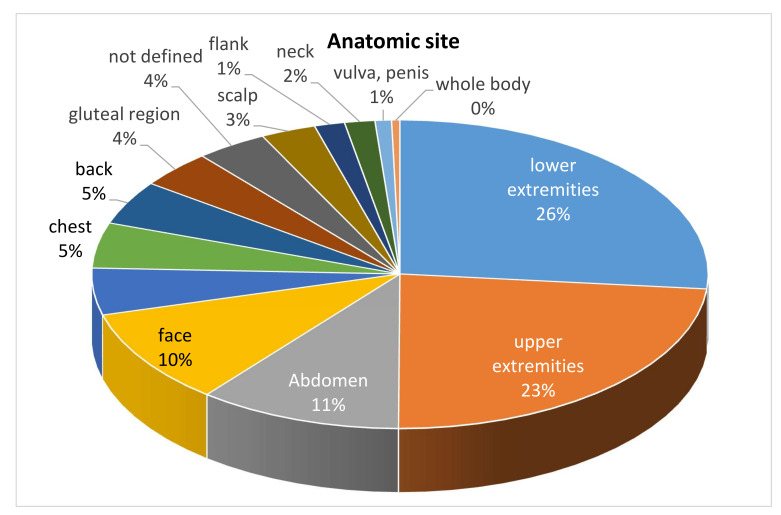
Anatomic site of infection.

**Figure 5 jof-08-00194-f005:**
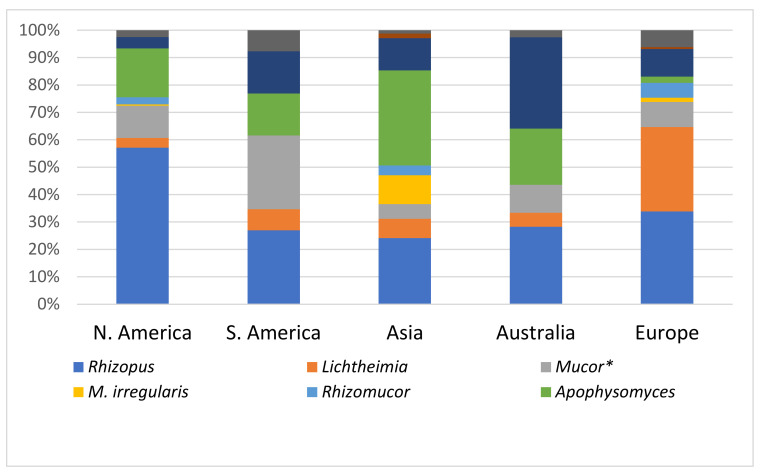
Geographic distribution of causative Mucorales organisms. Two cases were from Tunisia, Africa, and are not included in the Figure. * All *Mucor* species except *Mucor irregularis*.

**Table 1 jof-08-00194-t001:** Underlying diseases/conditions predisposing to mucormycosis.

Underlying Disease/Condition	Cases	
	n	% ^a^
Diabetes mellitus	139	20
Hematological malignancy	109	15.7
Acute myeloid leukemia	42	
Acute lymphoblastic leukemia	34	
Chronic myeloid leukemia	5	
Chronic lymphocytic leukemia	5	
Myelodysplastic syndrome	9	
Lymphoma	13	
Multiple myeloma	1	
Aplastic anemia	8	1
Allogeneic transplantation	28	4
Solid organ malignancy	7	1
Lung	1	
Brain	2	
Osteosarcoma	1	
Renal	1	
Prostate	1	
Colon	1	
Solid organ transplantation	57	8.2
Renal	22	
Liver	21	
Heart	9	
Lung	2	
Multi-visceral	3	
Neutropenia	45	6.4
Corticosteroids	59	8.5
Autoimmune disease	23	3.3
Systemic lupus erythematosus	7	
Inflammatory bowel disease	4	
Rheumatoid arthritis	3	
Psoriasis	2	
Other ^b^	7	
Renal failure	22	3.1
HIV	11	1.5
Prematurity	20	2.8
Extreme low birth weight	17	2.4
Cirrhosis	7	1
Alcoholism	6	0.8
Malnutrition	3	0.4
Other ^c^	10	1.4
None	275	39.6

^a^ The sum of the underlying diseases is greater than 100% because, in many cases, there were more than one. ^b^ Other includes two cases of angiitis, two of pemphigoid, one of autoimmune hepatitis, one autoimmune hemolytic anemia, and one idiopathic thrombocytopenic purpura. ^c^ Other includes two cases of beta-thalassemia, one case of CARD9 deficiency, two cases of hyposplenism, two of toxic epidermal necrolysis, one case of malabsorption syndrome, one of multiorgan failure, one Pearson syndrome (deferoxamine treatment), and one COVID-19 infection.

**Table 2 jof-08-00194-t002:** Modes of transmission.

Mode of Transmission	Cases	
	n	%
Major trauma	235	33.9
Motor vehicle accident (MVA)	104	
Surgery	47	
Burn	33	
Natural disaster	22	
Crush injury	8	
War injuries	6	
Fall	12	
Farm accidents	3	
Minor trauma	117	16.9
Injection sites	41	
Insect/arthropod bites	16	
Animal bites/scratches/kicks	8	
Gardening/plants	22	
Other minor injury	30	
Other trauma	23	3.3
Healthcare associated	108	15.6
Adhesive tapes/bandages	37	
Catheter insertion sites	46	
Hospital linen	8	
Herbal dressings on trauma	3	
Karaya ostomy bags	2	
Thoracic, abdominal drains	3	
Wooden tongue depressors	3	
Other nosocomial material	6	
Immersion in freshwater	8	1
Unknown	203	29.3

**Table 3 jof-08-00194-t003:** Isolated fungi.

Organisms	Cases	Mortality
	*n* (%)	*n* (%)
*Apophysomyces*	109 (18)	46 (42)
*A. elegans*	*58*	
*A. mexicanus*	*1*	
*A. trapeziformis*	*20*	
*A. variabilis*	*30*	
*Cunninghamella*	18 (3)	12 (67)
*C. bertholletiae*	*13*	
*Cunninghamella* spp.	*5*	
*Lichtheimia*	61 (10)	12 (20)
*L. corymbifera*	*45*	
*L. ramosa*	*7*	
*Lichtheimia* spp.	*9*	
*Mucor*	75 (12)	20 (27)
*M. circinelloides*	*12*	
*M. hiemalis*	*5*	
*M. indicus*	*2*	
*M. irregularis*	*21*	
*M. pusillus*	*3*	
*M. ramosissimus*	*1*	
*M. velutinosus*	*1*	
*Mucor* spp.	*30*	
*Rhizomucor*	18 (3)	4 (22)
*R. pusillus*	*4*	
*R. variabilis*	*4*	
*Rhizomucor* spp.	*10*	
*Rhizopus*	209 (34)	78 (37)
*R. arrhizus (oryzae)*	*78*	
*R. azygosporus*	*1*	
*R. delemar*	*3*	
*R. homothallicus*	*1*	
*R. microsporus*	*24*	
*R. microsporus var. rhizopodiformis*	*15*	
*R. oligosporus*	*1*	
*R. pusillus*	*1*	
*R. stolonifer*	*2*	
*Rhizopus* spp.	*83*	
*Saksenaea*	59 (10)	8 (13)
*S. vasiformis*	*49*	
*S. erythrospora*	*9*	
*Saksenaea* spp.	*1*	
*Syncephalastrum*	4 (1)	1 (25)
*S. racemosum*	*1*	
*Syncephalastrum* spp.	*3*	
Unidentified *Mucorales*	52 (9)	22 (42)
Total	605	203 (33)

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
