# Peer review of "Global Cutaneous Mucormycosis: A Systematic Review"

_jof, 2022, doi:10.3390/jof8020194_

Round 1
Reviewer 1 Report
The work of Skiada and cols. is an interesting systematic review of the epidemiology, clinical presentation, diagnosis and treatment of cutaneous mucormycosis. In general, the methodological strategy was adequate, as well as its data analysis and discussion were properly addressed. Moreover, the manuscript is well structured and written. The work is informative and contributes to the current state of the art of mucormycosis disease.
Author Response
Dear reviewer,
Thank you for your comments. It seems that you don't want to change anything in the manuscript, and I really appreciate this.
Sincerely
Anna Skiada
Reviewer 2 Report
The systematic review of world-wide cutaneous zygomycosis is the large analysis of the recent epidemiology data, diagnosis, and treatment. The review could have focused more on a more detailed analysis and description of patient populations in a meta-analysis of recent study data, however, even this simpler view will be valuable for an epidemiological overview of the increasing incidence of cutaneous zygomycosis.
Minor revisions:
General remark or question:
- It is possible to calculate the real incidence and prevalence in population of cutaneous zygomcosis in single sentence?
- Figure 2: In the discussion, can you explain the increasing trend of cutaneous zygomycosis incidence during the years explicitly?
Line 118: please unify size and font of the letters in the text
Line 122: The table is graphically confusing, adjust the alignment and, for example, add dividing lines in the header, etc.
Line 169: Same as before, rework the table for better quality.
Line 256: Unify rowing and design with other tables
Author Response
Thank you for reviewing our manuscript. The adjusted tables are shown on the revised manuscript.
Response to Reviewer 2 Comments
- Point 1: It is possible to calculate the real incidence and prevalence in population of cutaneous zygomcosis in single sentence?
Response 1: We have inserted two sentences in the first paragraph of the discussion: “There are publications reporting increased incidence of mucormycosis in general, as a result either of increased prevalence of diabetes, especially in Asia, or of new treatments of malignancies and autoimmune diseases [4, 5]. However, it is not possible to estimate the exact incidence or prevalence of cutaneous mucormycosis, because most data are from case reports or case series.” This is the response to points 1 and 2.
- Point 2: Figure 2:In the discussion, can you explain the increasing trend of cutaneous zygomycosis incidence during the years explicitly?
Response 2: Please see the response to point 1. In addition, may I draw your attention to the first paragraph of the discussion where we have already written that “There are multiple factors leading to increased numbers of mucormycosis in a country. There may be actual high incidence, or an increased rate of recognition because of better awareness, expertise and competence in mycological diagnosis [45]. From 1958 to 2000, there were only 8 cases of cutaneous mucormycosis published from India, and 134 from the USA. As the diagnostic facilities in healthcare centers of developing countries are improving, the number of cases reported, especially from India, are increasing alarmingly.”
- Point 3: Line 118: please unify size and font of the letters in the text
Response 3: Thank you for pointing out our mistake. We have corrected it.
- Point 4: Line 122: The table is graphically confusing, adjust the alignment and, for example, add dividing lines in the header, etc.
Response 4: We have adjusted the alignment in the table, and, it is now hopefully more clear.
- Point 5: Line 169: Same as before, rework the table for better quality.
Response 5: We improved the quality of the table, as suggested.
- Point 6: Line 256: Unify rowing and design with other tables
Response 6: We improved the quality of the table, as suggested.